# A Deep Learning-Based Password Security Evaluation Model

**Ki Hyeon Hong** [1] and **Byung Mun Lee** [2,*]

1   Department of IT Convergence, Gachon University, Seongnam-si 13120, Korea; yjs03075@gachon.ac.kr
2   Department of Computer Engineering, Gachon University, Seongnam-si 13120, Korea
*   Correspondence: bmlee@gachon.ac.kr; Tel.: +82-31-750-4756

**Abstract:** It is very important to consider whether a password has been leaked, because security can no longer be guaranteed for passwords exposed to attackers. However, most existing password security evaluation methods do not consider the leakage of the password. Even if leakage is considered, a process of collecting, storing, and verifying a huge number of leaked passwords is required, which is not practical in low-performance devices such as IoT devices. Therefore, we propose another approach in this paper using a deep learning model. A password list was made for the proposed model by randomly extracting 133,447 words from a total of seven dictionaries, including Wikipedia and Korean-language dictionaries. After that, a deep learning model was created by using the three pieces of feature data that were extracted from the password list, as well as a label for the leakage. After creating an evaluation model in a lightweight file, it can be stored in a low-performance device and is suitable to predict and evaluate the security strength of a password in a device. To check the performance of the model, an accuracy evaluation experiment was conducted to predict the possibility of leakage. As a result, a prediction accuracy of 95.74% was verified for the proposed model.

**Keywords:** authentication; deep learning model; information security; IoT; password security strength

## 1. Introduction

The use of passwords is an important authentication method to protect user information in the field of information security [1,2]. When users choose passwords, security and convenience mutually conflict [3,4]. For example, while a password that is composed in a complicated manner by combining uppercase and lowercase letters, numbers, and special symbols, such as "Qw3@lv53", is safe since it cannot be easily inferred by other users, it is inconvenient to use because it cannot be easily remembered by the user. On the contrary, while a password that combines regular patterns or easily inferred words, such as "abcdefg123", can be easily remembered by the user, it cannot perform the authentication function properly due to low security. Therefore, convenience and security must be appropriately considered when users select passwords, and websites or systems that provide the existing authentication functions should evaluate the security of the passwords that are selected by the users and improve security with appropriate feedback [5,6].

Currently, the most frequently used password security evaluation method is the evaluation indicator presented by the National Institute of Standards and Technology (NIST). To evaluate security, it analyzes the complexity of passwords according to the combinations formed by uppercase and lowercase letters, numbers, and special symbols [7]. In addition, a study on password security by Concordia University presented the zxcvbn evaluation indicator, which evaluates security by categorizing a list of passwords that are frequently used by general users [8]. However, existing password security evaluation indicators like these do not take the leakage of passwords into account. A password attacker can select various attack models [9]. For example, a dictionary attack is a method in which cracking is attempted by building a dictionary of letter strings that are likely to be selected as passwords [10,11]. While existing leaked password lists are good data for creating dictionaries, they can be seen as being vulnerable to dictionary attacks. In this respect, even

if the complexity of a password that is selected by a user is high, an attacker can easily crack it if the password has already been leaked. Therefore, whether a password has actually been leaked must also be considered to evaluate password security reliably [12].

However, to verify whether there has been a leak, vast amounts of leaked password data must be collected and stored. This is problematic for low-performance computing environments or terminals since they lack the memory capacity, and handling such a large volume of data will result in a performance drop. This method also has a structural weakness because password attackers can analyze the database that is stored in one place to determine the password selection tendencies of the users [13]. In addition, if a new leaked password occurs, there is the inconvenience of having to manage it. To solve this problem, a new password security prediction method that can determine whether a password has been leaked is needed. Among them, the deep learning-based evaluation model learns the weights of the model by using the features that are extracted from the training data, and the learned weights are used to predict the results of new data that were not used during training [14,15]. In particular, since data used for training is not required in the prediction process, there is no need to collect or store vast amounts of leaked password data, and the same level of security can be guaranteed in a low-performance computing environment. In addition, when only the trained model is used, there will not be any significant effect even if the performance of the processor is low. Furthermore, it is impossible for an attacker to analyze the password selection tendencies of the users with the learned weight.

Therefore, this paper proposes a security evaluation model that predicts whether a password selected by a user has been leaked by using deep learning. This model is divided into a process in which the model learns by collecting the features of passwords and whether they have actually been leaked, and a process of predicting whether a password selected by a user has been leaked. The feature data for training are the security evaluation scores that were extracted from existing password security evaluation indicators, and information about whether the passwords were leaked is obtained from the leaked password data. The weight of the trained model is used to predict whether the password selected by a user has been leaked. The reliability of the security evaluation model is confirmed through an accuracy evaluation experiment in which the model predicts whether there has been a leak, and this is compared with the results of predicting leakage with existing password security evaluation indicators.

In Section 2 of this paper, existing password security evaluation methods are considered, and deep learning-based evaluation models are examined. Section 3 presents a deep learning-based password security evaluation model and defines the feature data and label data for learning, and the learning method. In Section 4, experiments to verify the accuracy of the evaluation model are carried out and the results are analyzed. The conclusion is drawn in Section 5, the last section.

## 2. Related Studies

### 2.1. The Cracking Process and Method of the Password Attacker

In a typical system, a password selected by a user is converted into a hash value through a hashing algorithm and then stored [16]. Afterwards, when the user inputs the same password, an authentication process is performed to verify whether the user is a registered user. If the hash value matches, it is approved, but if it does not match, authentication is rejected. Since an inverse function for the hashing algorithm does not exist, the password cannot be inferred from the converted hash value [17]. Therefore, a password attacker must attempt all cases to infer the user's password. Figure 1 shows the brute-force attack process in which an attacker attempts all cases to crack a user's password. However, since the brute-force attack can take a lot of time as the password length becomes longer and the number of possibilities increases significantly, the attacker may attempt a variety of methods.

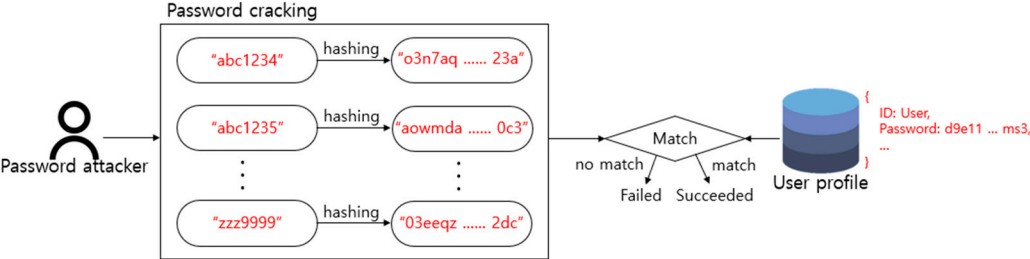

**Figure 1.** Brute-force password cracking process.

Since it is difficult to memorize passwords that are composed completely randomly, users tend to set passwords by using words in the dictionary and adding or deleting several letters, numbers, and special symbols [18]. In some cases, individual users may reuse previously used passwords, and they may also select passwords based on personally identifiable information (PII) using keywords that only they know, or their date of birth [19,20]. Since passwords like these are easier to predict than randomly composed letter strings, they can lead to a decrease in security. Therefore, an attacker may exploit this by building a list of passwords a user frequently uses, and attempt cracking based on this [21,22]. This is the targeted guessing attack, and existing leaked passwords can be abused by attackers to create user-specific password lists [20].

Ultimately, since password attackers can attempt cracking by employing a variety of methods, selecting a password directly after becoming fully familiar with the security guidelines for passwords is not an easy task for users [23]. Therefore, a password evaluation indicator that can evaluate how safe a password is in advance is needed.

### 2.2. Existing Security Evaluation Methods and Their Limitations

For the password security evaluation indicator, security must be evaluated by considering the password selection tendencies of the user, and this is carried out as shown in Figure 2. Here, the security evaluation score of the password selected by the user is predicted through addition and deduction indicators for security. For example, the most frequently used method at present is lower and uppercase letters, digits, and symbols (LUDS), and it is a method in which the requirement for the number of times letters, numbers, and special symbols are used is reflected. Indicators such as password length and variety of combinations belong to addition indicators since they make the password more complicated, and patterns in which letters continue sequentially or repeat, such as "AAA" and "123", belong to deduction indicators since they make the password simple [7].

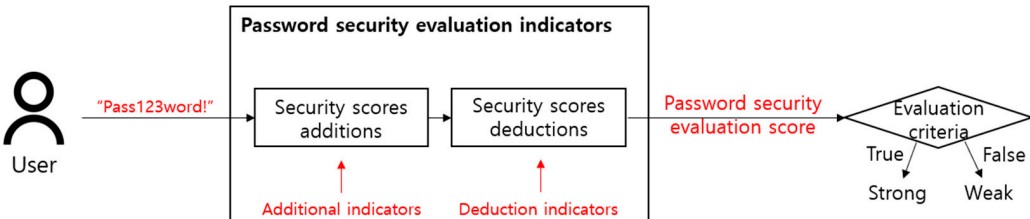

**Figure 2.** Existing password security evaluation process.

Like this, the password security evaluation score drawn from the password security evaluation indicators in Figure 2 can be classified into a password with strong security and a password with weak security by determining whether a certain standard is satisfied. For example, the password "Pass123word!" is classified as a password with strong security from the fact that it contains uppercase letters, lowercase letters, numbers, and special symbols and satisfies the LUDS requirement. In addition, if numbers or special symbols are put between words, as in "Pass123word!", it is classified as being very secure in zxcvbn by determining that there is no word that coincides with the words in the dictionary.

However, a security evaluation method like the one in Figure 2 does not check whether the password has been leaked no matter how high its security evaluation score was evaluated to be. In order to check whether a password has been leaked, it is necessary to collect and store existing leaked passwords and compare them with the password selected by the user. However, storing a leaked password list like this can be limiting in low-performance environments, and since new leaked passwords that occur must be added to the list, it is inconvenient. In addition, since the original string cannot be inferred from the hashed data, encrypting and storing the passwords is limiting for methods that evaluate the security of passwords based on the original letter strings such as zxcvbn or Levenshtein distance. In this regard, there is room for an attacker to analyze the stored leaked password list and abuse it to determine the password selection tendencies of the users. In order to solve this problem, a new evaluation method to determine whether a password has been leaked is needed.

### 2.3. Use Cases of Using Passwords in Low-Performance IoT Sensors

Sensor devices used in IoT services such as smart homes and smart healthcare transmit the data they collect and user information to a central server. In a smart home example like the one in Figure 3, data measured by a human detection sensor and gas detection sensor are transmitted to a server through a home gateway [24]. If gas is detected or an abnormal situation occurs when there is no one in the smart home, the gas valve must be closed automatically by the smart home gateway. Since the sensor data that was measured at the gateway must be transmitted between the gateway and the server in a secure session, authentication that can be mutually trusted is required. One of the methods is to use the password provided by the smart home user. When the gateway sends the hashed password to the server together with the sensor data, the server will identify the gateway and create a secure session using a password. Since session security is closely related to password security in this case, security evaluation of the password is important.

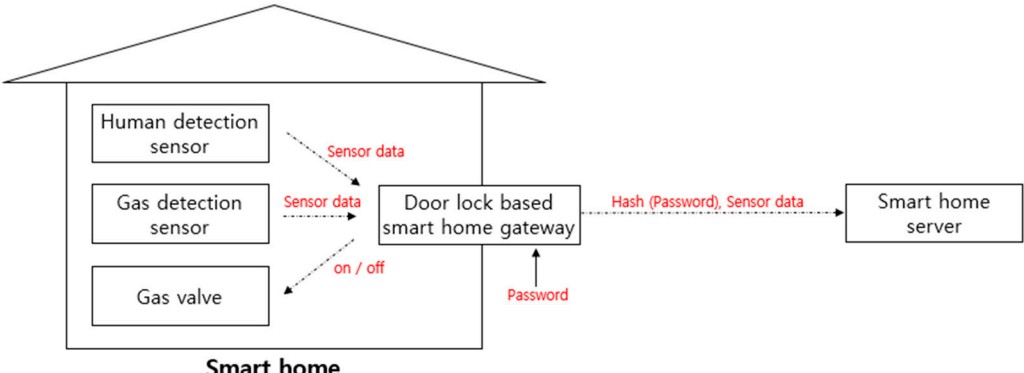

**Figure 3.** Low-performance IoT-based smart home service.

However, a problem can occur if the gateway has to send a password to the server to evaluate the security of the password. If a password attacker intercepts a message containing the password of a smart home gateway, a secure session between the gateway and the server cannot be guaranteed. In this respect, although the security of passwords must be evaluated at the gateway, there is a limit in building a leaked password database in a low-performance IoT environment. Therefore, a security evaluation model that takes this into consideration is needed.

### 2.4. Evaluation Model Using the Deep Learning Technique

Deep learning is a technique that is mainly used to predict results from feature data by using ANN (artificial neural networks) [25]. ANN process information by imitating the learning process of the human brain, and they are implemented by stacking perceptrons in multiple layers so that they can function like neurons [26]. For learning, the weights of the

perceptron are updated by using the differential value of the optimization function and the loss function to find the minimum over the entire region [27]. This process is learned in a hidden layer, and it is useful for solving problems that require predicting results for new data by finding and generalizing the patterns in the training data. The result learned by using artificial neural networks is also referred to as an evaluation model. The evaluation model is created during the training process and is used during the prediction process, as shown in Figure 4.

**Figure 4.** Deep learning training, and prediction process: (**a**) training process for a model; (**b**) prediction process using a model.

Figure 4a shows the process of training a model by extracting features from the training data and the validation data, and it is divided into a data preprocessing process and model training process. The data preprocessing process extracts predefined feature points from the data, and these are used for model training through normalization. The model training process consists of model training and model evaluation, and the weights and structure of the evaluation model are stored. These are used in the model predicting process shown in Figure 4b. After extracting features from data that were not used for training, normalization is carried out. At this point, a result is predicted by using the weights and structure of the evaluation model, which uses the features extracted from the training data. The predicted results are classified according to the classification criteria and output as the final prediction result.

Since a large amount of training data is required in the learning process like the one shown in Figure 4a, there is a limit to carrying it out in an IoT environment with low storage capacity and processing performance. Therefore, it is more appropriate to perform the learning process of the evaluation model in a high-performance computing environment. Additionally, when using the trained model that is generated as a result of learning, it can be separated from the learning process [28], as shown in Figure 4b. If this method is used, the concealment of the training data is guaranteed and the limitation of the IoT device can also be solved.

In particular, since the weight of the trained hidden layer is a numerical matrix that cannot be interpreted, it is impossible to infer from the original data [29]. The weights of the artificial neural network are initialized randomly, and training is carried out by reducing the error between the prediction result of the training data and the label data, which is the result that must be predicted. Therefore, depending on the random initialized weights, different weights can result even from the same training data, and it can be seen as converging a vast amount of training data into relatively few weights. In this respect, the weights of the artificial neural networks guarantee the concealment of the result prediction, and it

is also appropriate to use the evaluation model in low-performance environments such as browsers and small IoT devices. Therefore, a security evaluation model that predicts whether a password selected by a user has been leaked by using deep learning will be proposed in this paper.

## 3. Methodology

The proposed deep learning-based password security evaluation model predicts whether passwords have been leaked by using artificial neural networks. This is divided into the process of training a model on a server with low performance restrictions, as shown in Figure 5a, and the process of predicting leakage from a low-performance IoT device by using the trained evaluation model as shown in Figure 5b. Figure 5 is largely classified into four phases consisting of data selection, feature extraction for model training, evaluation model training, and leaked password prediction.

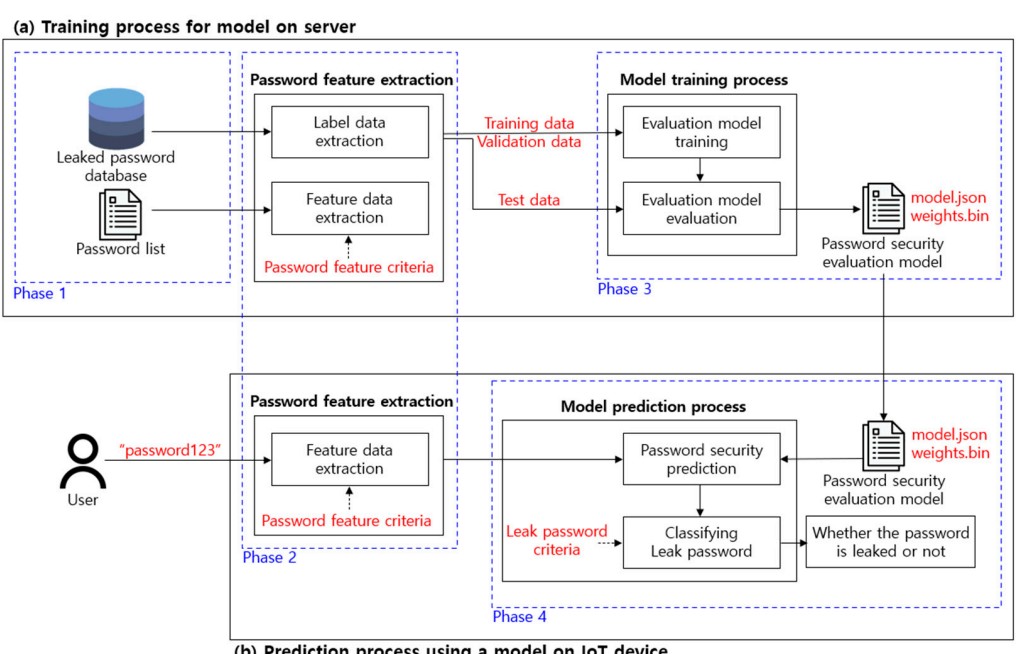

**Figure 5.** Phases of the proposed method: (**a**) training process for model on server; (**b**) prediction process using a model on IoT device.

In Phase 1 of Figure 5, a list of passwords for training the evaluation model is collected and a leaked password database is defined. In Phase 2, feature data and label data are defined and extracted for training the evaluation model. The point to consider at this time is that it must be possible to extract the feature data identically for all password lists, and each feature must not have any associations. In Phase 3, the password security evaluation model is built and trained by using a deep learning-based artificial neural network model. In order to select the evaluation model that was built, the model is trained by classifying the data into training data, validation data, and test data and the performance is checked. In the last phase, the leaked password criterion is defined to predict whether the password selected by the user has been leaked, and the result is classified and returned.

Since Figure 5a,b are structurally separated, Figure 5b no longer requires a password list for training the evaluation model. By doing this, it is possible to prevent a password attacker from inferring a user's password that had been used for the training by using an IoT device. Additionally, since the weights of the evaluation model that is already trained are used in Figure 5b, it is a model that is appropriate for low-performance environments.

In addition, the weight of Figure 5a may be received periodically to update the evaluation model. For example, even if the weight (weights.bin) of the evaluation model (model.json) is leaked to the password attacker in Figure 5b, the updated evaluation model

can be requested and received from Figure 5a. This updated evaluation model continuously learns the weight from Figure 5a each time a password leak occurs. In this case, since the initial weight of the learned weight is different from that of the training data, a model with a completely different weight than the evaluation model that was leaked to the attacker is learned as the result. Therefore, Figure 5b can receive the weight of this evaluation model and evaluate the password security with the model with a weight that is completely different from the leaked evaluation model.

### 3.1. Phase 1: Data Source

### 3.1.1. Password List

First, the password list is extracted that will be used as training data as the primitive password data before the feature data for training the evaluation model. The passwords included in the password list should be composed by considering the actual password selection tendencies of the user. However, it is illegal to collect the actual passwords of users and build lists, and there is potential for an adverse security impact. To solve this problem, the password list is built with a two-step process as shown in Figure 6.

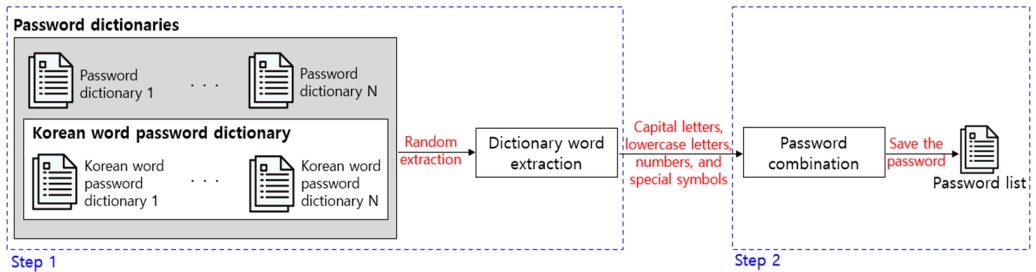

**Figure 6.** Process of creating a password list.

Step 1 in Figure 6 is a process of randomly extracting words after building dictionaries with words that users frequently use as passwords. The password dictionary includes a list of frequently used English words in Wikipedia, a list of American movies and TV programs, a list of password statistics frequently used by users, a list of first and last names frequently used by Americans, and a list of keyboard patterns that are easy to remember. This was obtained from zxcvbn, an evaluation indicator that evaluates the password security of actual English-speaking users [23]. However, zxcvbn does not consider the password selection tendencies of non-English language speaking users. To solve this problem, a list of Korean nouns and a list of male and female names frequently used by Koreans were added in a password dictionary in this paper [30].

However, PII-based passwords are not considered in this study. PII-based passwords vary for each individual, and there is a risk of personal information leakage if they are built into a dictionary. In addition, since the extracted feature data are different, an evaluation model with different weights for each individual will be trained. Therefore, the password list does not consider targeted guessing, and it is limited to only trawling guessing from Korean-language-based password users among universal English-speaking and non-English-speaking users.

### 3.1.2. Leaked Password Database

The deep learning-based evaluation model performs supervised learning to learn the weights of the model by using the feature data and label data, which is the prediction result. Therefore, collecting whether the passwords have been leaked, which is the goal of the prediction, is important. However, it is very difficult to collect leaked passwords by obtaining passwords directly from existing users. Therefore, an existing leaked password database that is provided is imported and used instead [31]. Among them, "Have I Been Pwned" hashes and stores passwords that have been leaked by attackers from various

websites around the world [31]. If this is used, the number of times a password selected by a user has been leaked can be easily verified, and whether it has been leaked is labeled.

It is impossible to obtain the password of the leaked original from the hashed leaked password database. As shown in Figure 6, the password in the password list is hashed, and whether the corresponding password matches the hashed value of the leaked password database is checked. If there is a matching hash value, it is labeled as a leaked password, and if there is no matching hash value, it is labeled as a password that has not been leaked.

*3.2. Phase 2: Password Feature Extraction*

The feature data and label data are extracted by using the password list and the leaked password database. This is as shown in Figure 7. Step 1 in Figure 7 is the process of generating a password feature matrix by using the password list. Each data in the password list acquires a feature score according to a defined feature criterion. It is composed of a matrix, and it is used as feature data for model training. For example, "password 123" in Figure 7 has only lowercase letters and numbers among uppercase letters, lowercase letters, numbers, and special symbols, and the feature score is 1 because the letters and numbers were not mixed and used. In addition, the feature score of zxcvbn is zero because it used the word "password" that is included in the password dictionary and the consecutive numbers "123". Finally, since the Levenshtein distance with the original letter string "password" is 3 because the number "123" is included, the feature score is 1.

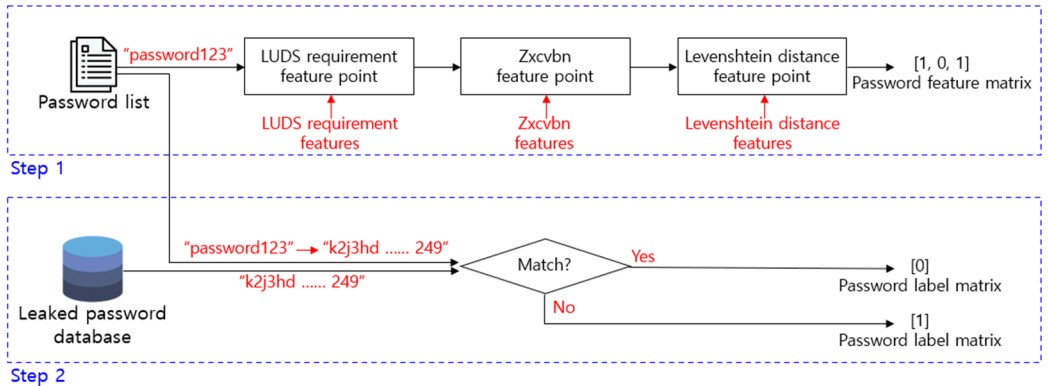

**Figure 7.** Process of extracting password features.

Step 2 is the process of extracting the label data, which is the prediction target of the evaluation model, by using the password list and the leaked password database. The password from which the feature data has been extracted should be composed as one data set with the label data, and just like the feature data, there should be no empty value. Since the leaked password database has been hashed, whether there is a hash value that matches is verified by hashing the password from which features have been extracted. If there is a matching hash value, it is labeled as a leaked password, and if there is no matching hash value, it is labeled as a password that has not been leaked. This is composed as a password label matrix, and it is a prediction result with a value of 0 or 1.

3.2.1. Step 1: Feature Data Extraction

The feature data should have no empty values for all passwords in the password list. In addition, each feature should not be related to each other since learning may become biased if each feature is related. In this paper, existing password security evaluation indicators are used to extract features. This is shown in Table 1. For example, the LUDS requirement evaluates security with a combination of uppercase letters, lowercase letters, numbers, and special symbols, and zxcvbn evaluates security based on frequently used passwords. In addition, Levenshtein distance evaluates security based on the number of letters that are different between the original letter string and the selected letter string.

In this respect, it can be seen that each password security evaluation indicator evaluates security by analyzing one password with a different criterion.

**Table 1.** Password feature data list.

| Feature | Description | Score |
|---|---|---|
| LUDS requirement | Combination of uppercase letters, lowercase letters, numbers, and special symbols | 0–4 |
| zxcvbn point | List of frequently used passwords | 0–4 |
| Levenshtein distance | Difference between the original letter string and the comparison letter string | 0–1 |

The security evaluation score that is extracted with the security evaluation indicators is converted into a numeric form to be used as input data for the evaluation model. If Table 1 is examined, the LUDS requirement and zxcvbn are expressed in five steps, from 0 to 4, and the Levenshtein distance is expressed in two steps of 0 and 1. This follows the security evaluation classification criterion proposed by each security evaluation indicator. For example, the LUDS requirement classifies the security evaluation scores into five steps by using the addition and deduction indicators. zxcvbn measures the complexity of combining dictionary words with uppercase letters, lowercase letters, numbers, and special symbols and classifies them into five steps. Since Levenshtein distance is defined as the distance between two letter strings, it has a value of 0 or higher. However, if a large value is reflected as it is without a threshold in the model training process, distorted results may be obtained by being sharply biased to one side and affecting other values. Therefore, preprocessing that normalizes the values of the Levenshtein distance is needed. Since a password with a distance of 3 or more was classified as having strong security in an existing example that studied the security of passwords by using the Levenshtein distance, in this paper, a distance less than 3 is defined to be 0 and a distance of 3 or more is defined to be 1 [32,33].

### 3.2.2. Step 2: Label Data Extraction

The password security evaluation model predicts whether the password selected by a user has been leaked. Therefore, it is classified into either a password that has been leaked or a password that has not been leaked, and the criteria are shown in Table 2. According to the leaked password database, for example, "password123" in Figure 7 is a password with very low security, with a hash value that has been leaked 126,927 times [31]. Label data for this can be seen as 0 according to Table 2. In contrast, since the password "Passwordq123" does not have a matching hash value, the label data is set to 1.

**Table 2.** Password label data list.

| Label | Description |
|---|---|
| 0 | A password that has been leaked one or more times |
| 1 | A password that has not been leaked |

Since the collected feature data and label data are simple numerical matrices, it is impossible to infer the original password before extraction. Therefore, even if a password attacker steals the password matrix that is used for training, the user's password cannot be inferred, and there is no need to store the original passwords for training.

### 3.3. Phase 3: Building a Password Evaluation Model with Deep Learning

A deep learning-based password security evaluation model is built by using the feature data of the password that was built with the above procedure and the label data that includes whether there has been a leak. The model is suitable for low-spec IoT environments, and it must be impossible for an attacker who has obtained the weights and

structures of the trained model to analyze any information. Therefore, a fully connected neural network, which is frequently used to predict results with a simple structure and a small number of operations, is selected. The fully connected neural network is composed of an input layer for inputting the feature data, a hidden layer for performing the actual prediction, and an output layer for outputting the prediction results, as shown in Figure 8.

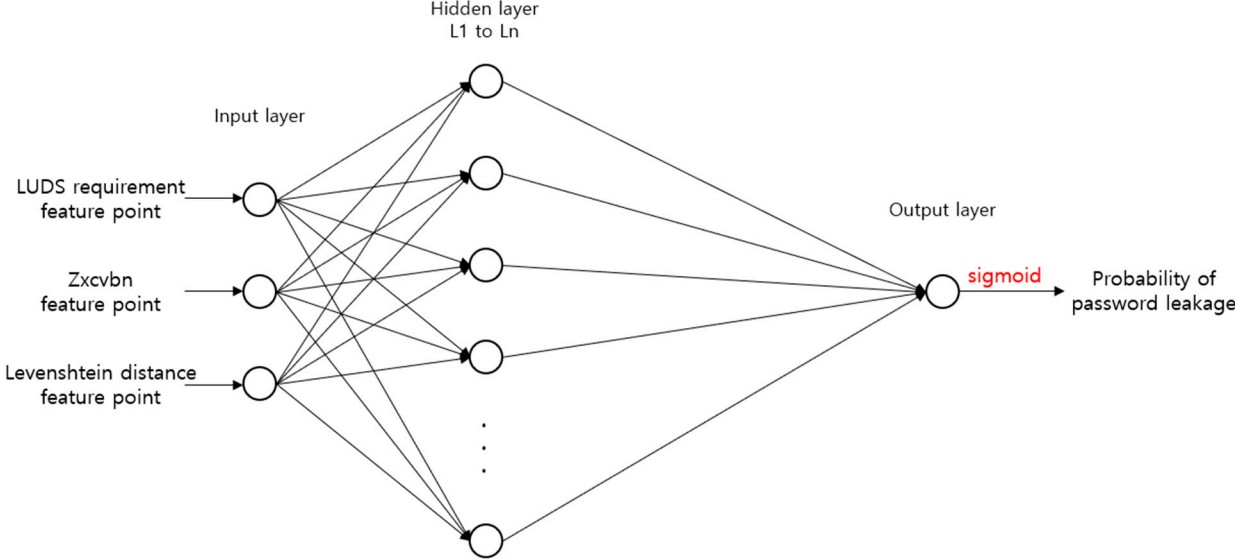

**Figure 8.** Structure of password evaluation model with deep learning.

The input layer of the password security evaluation model in Figure 8 receives the three pieces of feature data in Table 1 as input. They are used to learn the weights of the hidden layer, and the weights are updated by comparing the password leakage probability that was predicted as the result with whether the password has been leaked, which is the label data. For this, the prediction result of the evaluation model is normalized from 0 to 1, which is the range of the label data. This is solved by applying the sigmoid function to the result that is returned to the output layer.

In general, the choice of hyperparameters such as the depth, activation function, and optimization function of the hidden layer is important to train an excellent evaluation model with a fully connected neural network. In order to check the accuracy of the trained evaluation model and select a model, the feature data and label data are separated into 70% training data, 20% validation data, and 10% test data as shown in Table 3. In the model training process of Figure 5, the training data and validation data are used to update the weights of the evaluation model and to determine the degree of learning. The trained evaluation model uses the test data to check and determine whether the evaluation model is properly trained.

**Table 3.** Data separation for deep learning.

| Data | Separation (%) | Number of Data |
| --- | --- | --- |
| Training data | 70 | 93,415 |
| Validation data | 20 | 26,688 |
| Test data | 10 | 13,344 |
| Total | 100 | 133,447 |

To this end, 68,582 passwords that have been leaked and 64,865 passwords that have not been leaked were obtained with the processes in Figures 6 and 7. Based on Table 3, they were classified into 93,415 training data, 26,688 validation data, and 13,344 test data.

*3.4. Phase 4: Classification of Predicted Results*

The prediction result that was preprocessed through the sigmoid function of Figure 8 is the probability of password leakage between 0 and 1. A password that has been leaked will be closer to 0, and a password that has not been leaked will be closer to 1. The criterion for a password that has been leaked is a real number value between 0 and 1, and it is important data that determines whether the password has been leaked. For example, if the leaked password criterion is 0.1, most passwords are classified as not having been leaked. On the contrary, if the leaked password criterion is 0.9, most passwords will be classified having been leaked. Therefore, it is necessary to determine the criterion for the passwords that have been leaked by evaluating the evaluation model that has actually been trained.

The selection of the model structure and the hyperparameters and deciding the criterion for the leaked passwords like this are important factors for improving the performance of the model. This is verified through experiments, and the excellence of the additionally optimized evaluation model compared to the existing password security evaluation methods will be confirmed.

## 4. Experiment and Evaluation

### 4.1. Data Preprocessing and Experiment Setup

In order to confirm the performance of the proposed evaluation model, the password list is collected and preprocessed according to the process in Figure 9. Figure 9a generates a password based on a word dictionary as shown in Figure 6. The password generated at this time was created by the user by transforming a word from the dictionary according to the existing research [34]. Figure 9b is the result of extracting the features from the password list that were collected in Figure 9a and labeling them like the process in Figure 7. For example, since "q1w2e3r4" in the password list is included in the password dictionary that is frequently used by users, the evaluation score of zxcvbn is zero. However, since lowercase letters and numbers were used alternately, the security evaluation score is 3 because it satisfies the LUDS requirements, and the security evaluation score is 1 because it satisfies the Levenshtein distance since it is not included in the English word dictionary. In addition, if the corresponding feature is input to the evaluation model, the expected output is equal to 0 since it is a password that has been leaked. Figure 9c classifies the password matrix extracted from Figure 9b as shown in Table 3. This is 70% training data, 20% validation data, and 10% test data, and the entire process of Figure 9 was performed in advance for the experiment. There are no overlapping passwords among the training data, validation data, and test data in the experiment process. In addition, k-fold cross validation was used to prevent the evaluation model from learning while being biased toward specific training data. For example, after evenly dividing the entire data into 10 parts, 70% of them are used as training data, 20% are used as verification data, and 10% is used as test data. The training data and verification data are changed for each epoch. In each epoch, the average accuracy and the average loss value are defined as the accuracy and loss value of the evaluation model.

The trained model can use models such as ANN, decision tree, and random forest. However, the decision tree is not suitable for models that evaluate the security of passwords because the classification criterion of each feature is not concealed. In addition, the random forest, which gathers classification results by varying the combinations of features, is not suitable for training this evaluation model, which has defined the number of features to be 3. Therefore, in this evaluation model, ANN, which is more suitable for learning, was used.

The evaluation model is trained, and the performance is checked by using the collected and classified data as in the process of Figure 9. Hyperparameters such as the depth of the hidden layer, the number of nodes, the activation function, and the optimization function in the learning process are very important factors because they directly affect the performance of the model. Therefore, they were defined as in Table 4 for the selection of the hyperparameters.

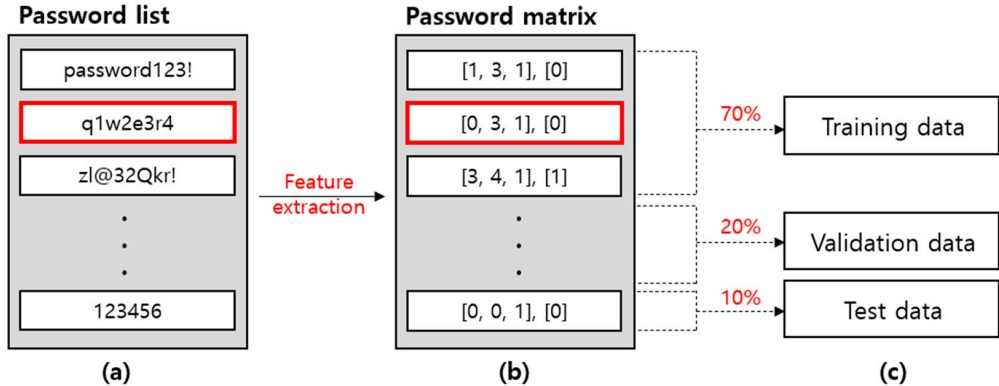

**Figure 9.** Process of preprocessing the password list and example (red box): (**a**) password list; (**b**) password matrix; (**c**) classification of data.

**Table 4.** Hyperparameter list.

| Hyperparameter | Description | Element |
|---|---|---|
| Depths | Depth of the hidden layer | 1, 2, 3, 4, 5 |
| Number of nodes | Number of nodes in each hidden layer | 1, 2, 4, 8, 16, 32 |
| Activation Function | Activation function of the hidden layer | ReLu |
| Optimizer | Optimizer for backpropagation | Adam |
| Dropout | Dropout of the hidden layer | 0 |

The depth of the hidden layer in Table 4 was limited from 1 to 5 to train models that are suitable for low-performance environments. In addition, 1, 2, 4, 8, 16, and 32 nodes were used in each hidden layer. Additional factors for training the model include an activation function, an optimization function, and a dropout. For the activation function, "ReLu" was used. "ReLu" outputs 0 when the input data is negative and 1 when it is positive, and it has an advantage in training speed due to the simple calculation. "ReLu" may lose the learned weight when the hidden layer becomes deeper, or the input data is 0 because differentiation becomes impossible. However, since the depth of the hidden layer was limited to 5 in the training process for this model, the risk of this is low. "Adam" was used as the optimization function. Dropout was not used because a small number of nodes were used in each hidden layer.

*4.2. Comparison with Previous Method and Model Selection*

The performance of the evaluation models that were trained by using the hyperparameters in Table 4 was compared and analyzed. To this end, after classifying the trained models according to the depth of the hidden layer, the average of the learning times, loss values, and classification accuracy were compared. To train the evaluation model, a desktop computer loaded with Windows 10, with an Intel(R) Core (TM) i7-10700 CPU, 16 GB of memory, and an NVIDIA GeForce GTX 1660 graphics card GPU was used. For the learning environment platform, @tensorflow/tfjs 3.7.0 was used in Node.js 14.15.3. As shown in Table 5, the training time, loss value, and accuracy that were measured in the training process experiment were obtained.

**Table 5.** Performance table of the evaluation model according to the depth of the hidden layer.

| Depths | Time (ms) | Loss (MSE) | Accuracy (%) |
|---|---|---|---|
| 1 | 58,692.666 | 0.076 | 87.443 |
| 2 | 89,923.305 | 0.085 | 85.91 |
| 3 | 116,020.203 | 0.089 | 84.877 |
| 4 | 164,500.302 | 0.109 | 80.587 |
| 5 | 185,474.527 | 0.179 | 65.651 |

The training time in Table 5 is the time to execute 20 epochs. The loss value and accuracy of the evaluation model were calculated with the loss value and accuracy of the verification data by considering the overfitting that occurs during the training process. This is the average value of the k-fold cross-validation result that was performed in each epoch. It was the shortest with an average of 58,692.666 ms when the depth of the hidden layer was 1, and the training time increased as the depth of the hidden layer increased. In addition, it was confirmed that the smaller depth of the hidden layer resulted in smaller loss values in the training process. Accuracy is the accuracy of classifying passwords that have been leaked as low security passwords and those that have not been leaked as high security passwords. If the accuracy in Table 5 is examined, the classification accuracy was the highest when the classification criterion was 0.5, and the best average classification accuracy, 87.443%, was shown when the depth of the hidden layer was 1. Figure 10 displays the data of Table 5 in graph form.

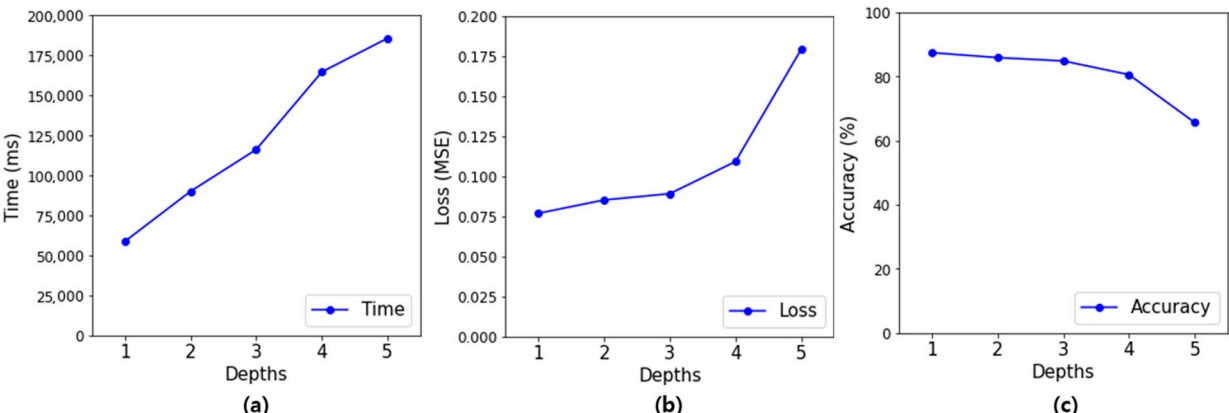

**Figure 10.** Performance graph of the evaluation model according to the depth of the hidden layer: (**a**) learning time according to the depth of the hidden layer; (**b**) loss according to the depth of the hidden layer; (**c**) accuracy according to the depth of the hidden layer.

If Figure 10a is examined, the training time tends to increase as the depth of the hidden layer increases. However, it was confirmed that an increase in training time did not positively affect the performance of the evaluation model and that the performance of the model decreased, as shown in Figure 10b for the loss value and Figure 10c for the accuracy. This is assumed to be due to excessive training during the model training process or because the weights became lost due to the deep hidden layers.

In order to determine the number of nodes for the best model in Figure 10 with a hidden layer depth of 1, the training time, loss value, and accuracy of the evaluation model were compared according to the number of nodes. They are as shown in Table 6, where the training time was the shortest, but the accuracy was confirmed to be 50% when the number of nodes was 1 because learning was not carried out. In addition, when the number of nodes was 16 and 32, the classification accuracy was the best at 94.937%. By considering the training time as well, the hyperparameters of the evaluation model were selected for the depth of the hidden layer of 1 and 16 as the number of nodes that are included in the hidden layer.

### 4.3. Experiment Results and Performance Evaluation

In order to verify the performance of the password security evaluation model that was selected earlier, the classification accuracy was compared with the existing password security evaluation indicators. The security evaluation score according to each security evaluation method and whether the passwords have actually been leaked were compared by using the test data of Table 3. They are as shown in Table 7.

**Table 6.** Performance table of the evaluation model according to the number of nodes.

| Number of Nodes | Time (ms) | Loss (MSE) | Accuracy (%) |
|---|---|---|---|
| 1 | 29,442 | 0.25 | 50 |
| 2 | 43,968 | 0.0423 | 94.934 |
| 4 | 55,314 | 0.0426 | 94.926 |
| 8 | 61,121 | 0.0422 | 94.926 |
| 16 | 78,198 | 0.0422 | 94.937 |
| 32 | 84,113 | 0.0422 | 94.937 |

**Table 7.** Comparison for password security evaluation performance.

| Index | Leak or Not | Security Evaluation Score | Count |
|---|---|---|---|
| Proposed evaluation model | Leaked password | Weak | 8351 |
| | | Strong | 179 |
| | Not leaked password | Weak | 389 |
| | | Strong | 4425 |
| Zxcvbn point | Leaked password | Very weak | 488 |
| | | Weak | 7611 |
| | | Average | 428 |
| | | Strong | 3 |
| | | Very strong | 0 |
| | Not leaked password | Very weak | 0 |
| | | Weak | 713 |
| | | Average | 657 |
| | | Strong | 784 |
| | | Very strong | 2660 |
| LUDS requirement | Leaked password | Very weak | 7643 |
| | | Weak | 751 |
| | | Average | 122 |
| | | Strong | 12 |
| | | Very strong | 2 |
| | Not leaked password | Very weak | 470 |
| | | Weak | 492 |
| | | Average | 791 |
| | | Strong | 617 |
| | | Very strong | 2444 |
| Levenshtein distance | Leaked password | Weak | 1484 |
| | | Strong | 7046 |
| | Not leaked password | Weak | 46 |
| | | Strong | 4768 |

The test data in Table 7 used 4814 passwords that have not been leaked and 8530 passwords that have been leaked. When the security of a password is evaluated, if a password that has been leaked is evaluated as a password that has not been leaked it may cause the user to mistake a password that has been leaked to have high security. Therefore, the number of passwords that were leaked was given a bit more consideration when the test data was collected.

In Table 7, the evaluation model proposed in this paper classified 8351 out of 8530 passwords that have been leaked as having weak security. It also classified 4425 out of 4814 passwords that have not been leaked as having strong security. This is classification accuracy of 95.74%. However, for Levenshtein distance in Table 7, only 1484 passwords that have been leaked were classified as weak security passwords, and 4768 of the passwords that have not been leaked were classified as ones with strong security. Since the classification

accuracy was 46.85%, it was confirmed that the Levenshtein distance had lower reliability compared to the proposed evaluation model.

However, it is difficult to directly compare the zxcvbn scores and LUDS requirements in Table 7 with the proposed evaluation model because they are not binary classification models that classify passwords as those with weak security and strong security. For example, the LUDS requirement classifies security into five steps from 0 to 4. At this time, if the criterion for the low security passwords is 1, the LUDS requirement in Table 7 classifies 8394 of the leaked passwords as low security passwords and 3852 of the passwords that have not been leaked as high-security passwords, with a classification accuracy of 91.77%. However, if the criterion is set to 2, it has a classification accuracy of 86.75%. To solve this problem, the recovery operation characteristic (ROC) curve was compared. The ROC curve is a method of evaluating the performance of the trained model by using the sensitivity to judge an answer as the correct answer and the specificity to judge an answer as the wrong answer. This is as shown in Figure 11.

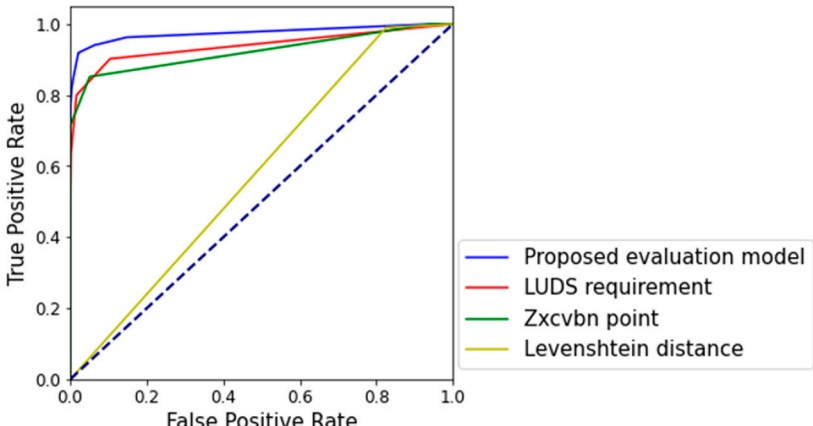

**Figure 11.** Password security evaluation ROC curve.

Figure 11 shows the ROC curves for the evaluation model proposed in this paper, the zxcvbn score, the LUDS requirements, and the security evaluation score of Levenshtein distance. Since the x and y axes are 1, the ROC curve has a maximum area of 1. It can be seen that the wider area of the ROC curve reflects better performance of the binary classification. The proposed evaluation model displayed the best binary classification performance at 0.975, and it was 0.938 for the LUDS requirement, 0.922 for the zxcvbn score, and 0.582 for the Levenshtein distance, which was verified to be the lowest binary classification performance.

Through this, it was confirmed that the proposed password security evaluation model evaluated the security of a password by considering whether the password has actually been leaked and that it was effective for dictionary attacks by password attackers. Additionally, since the evaluation model proposed in this study extracts features regardless of the password length or complexity and predicts whether a password was leaked, the time complexity is O (1). Therefore, the password security evaluation model is suitable for low-performance IoT devices.

## 5. Conclusions

In the field of information security, user authentication is an important factor for protecting personal information. Since passwords must be remembered by users, convenience and security must be appropriately guaranteed. However, the existing password security evaluation methods are very vulnerable to dictionary attacks by password attackers because they do not consider passwords that have been leaked. In addition, collecting and storing a vast quantity of leaked passwords puts limitations on using them in low-performance computing environments, and managing new leaked passwords is cumbersome.

Therefore, in this paper, a feature matrix was extracted by collecting passwords that have been leaked and those that have not been leaked. The feature matrix is a security evaluation score that is extracted by using the existing password security evaluation methods, and a security evaluation model that takes this as the input and predicts whether a password has been leaked was proposed. For performance evaluation, an experiment was conducted to compare the accuracy of classifying leaked passwords between the existing password security evaluation models and the proposed security evaluation model. As a result, it was confirmed that the proposed evaluation model had the highest classification accuracy. In addition, the evaluation model proposed in this paper structurally separates the training process and the security evaluation process so that password attackers cannot obtain any information by using the evaluation model, and it is suitable for evaluating security in low-performance environments.

However, personal information-based passwords or passwords reused by individual users were not considered in this study. In addition, since passwords that have been leaked more than once were classified as passwords with low security, passwords that have been leaked by accident were not processed. To improve this, it should be possible to add a password matrix that considers individual password selection trends by considering targeted guidance. In addition, three may be too small for the number of features to predict whether there was leakage. In this study, three small features were used to predict whether a password had been leaked in a low-performance IoT environment. By extracting a wider variety of password features, the accuracy of the proposed model can be increased. If this is supplemented, it will be possible to learn and provide a password security evaluation model suitable for individuals.

**Author Contributions:** K.H.H. and B.M.L. conceived and designed the experiments; K.H.H. performed the experiments; K.H.H. and B.M.L. analyzed the data; K.H.H. wrote the paper. K.H.H. and B.M.L. have read and approved the final manuscript. All authors have read and agreed to the published version of the manuscript.

**Funding:** This research was funded by the Ministry of SMEs and Startups grant number S2957039.

**Institutional Review Board Statement:** Not applicable.

**Informed Consent Statement:** Not applicable.

**Data Availability Statement:** The data presented in this study are available in https://github.com/KiHyeon-Hong/Password_security_test (accessed on 22 November 2021).

**Acknowledgments:** This work was supported by the Technology Development Program funded by the Ministry of SMEs and Startups (MSS, Korea) (Grant No. S2957039).

**Conflicts of Interest:** The authors declare no conflict of interest.

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
