# Peer review of "A Deep Learning-Based Password Security Evaluation Model"

_applsci, doi:10.3390/app12052404_

Round 1

Reviewer 1 Report

The work analyses and proposes a method to predict leaked password selection by the user using NN implementation. 

Conceptually, some explanation is needed to bother with such a piece of complex machinery to train/predict and so on, if one can protect IoT with entirely random passwords of length 11 where the password evaluation had a restriction to fail three times.

The work could be improved as follows:

  1. It would be very nice to emphasize IoT relationship to the whole storyline. It is not very clear what low-spec IoT devices will be protected. What are the minimum parameters for such devices to be able to predict leaked passwords on IoT device?
  2. The clarification in introduction and methodology of Deep-learning as a concept used in small power devices would be very welcome.
  3. Scenario that shows usage (use case, motivating example) of such trained model in early stages of the work and how the update of classifier is going to be made would be very welcome.
  4. In Section 3.1.2. the leaked password dataset is mentioned, but it is not very clear from text if passwords from [29] were already reversed from hash to normal strings.
  5. In the Model Levenshtein distance according the paper produces a score. In Table 1 score is denoted 0~1. What does the 0~1notation mean? Is it standard? If not change to something like values from the set {0, 1} as explained in text.
  6. When training ANN, have you considered overfitting? When does it happen? Some clarification on dataset split Training/Testing/Validation would help. Does the data subsets overlap?
  7. What are the CPU performance or complexity requirements for IoT to still be able to predict leaked password in a given time.
  8. Table 5 and 6 Time (ms) column does not make sense if there is no reference to device that experiments were performed on (for example targeted IoT device).

Author Response

Thank you for your review.
We revised what you pointed out in the manuscript and summarized how and what to revise.

The work analyses and proposes a method to predict leaked password selection by the user using NN implementation. 

Conceptually, some explanation is needed to bother with such a piece of complex machinery to train/predict and so on, if one can protect IoT with entirely random passwords of length 11 where the password evaluation had a restriction to fail three times.

The work could be improved as follows:

1. It would be very nice to emphasize IoT relationship to the whole storyline. It is not very clear what low-spec IoT devices will be protected. What are the minimum parameters for such devices to be able to predict leaked passwords on IoT device?

Yes, first of all, I would like to thank you for the review. As you pointed out, the explanation about the areas of the low-performance IoT environment to which the proposed evaluation model could be applied seems inadequate. By considering this, we added examples that can actually be used from line 152 to line 163 of Section 2.3. In addition, we added an explanation of why evaluating the password security in a low-performance IoT environment is important from line 164 to line 170.

2. The clarification in introduction and methodology of Deep-learning as a concept used in small power devices would be very welcome.

Yes, thank you for your good opinion. In order to reflect your suggestion, we added the method to introduce deep learning in a low-performance IoT environment, and an existing case as an example, from line 199 to line 206 of Section 2.4.

3. Scenario that shows usage (use case, motivating example) of such trained model in early stages of the work and how the update of classifier is going to be made would be very welcome.

Yes, thank you for your good opinion. In order to reflect your suggestion, we updated the evaluation model that was proposed in this study and added an applicable scenario from line 245 to line 254 of Section 3.

4. In Section 3.1.2. the leaked password dataset is mentioned, but it is not very clear from text if passwords from [29] were already reversed from hash to normal strings.

Yes, we revised the part that you pointed out. The leaked password database is in a hashed state. In order to use this, we composed a list of the words that users often use as passwords, hashed it, substituted the results to the database, and checked whether matching hash values existed. Since we felt that the explanation about this was insufficient, we added lines 292 to 297 in Section 3.1.2.

5. In the Model Levenshtein distance according the paper produces a score. In Table 1 score is denoted 0~1. What does the 0~1notation mean? Is it standard? If not change to something like values from the set {0, 1} as explained in text.

Yes, we revised the part that you pointed out. The Levenshtein distance is a natural number equal to or greater than 0, and there is no limit to the maximum value. This can have an adverse effect on deep learning in which learning becomes biased towards the corresponding feature value. Therefore, it was normalized to a value between 0 and 1, and it was normalized to the standard that was suggested in the password security study using the existing Levenshtein distance. Since we felt that the explanation about this was insufficient, we reflected this from lines 341 to 345 in Section 3.2.1.

6. When training ANN, have you considered overfitting? When does it happen? Some clarification on dataset split Training/Testing/Validation would help. Does the data subsets overlap?

Yes, we revised the part that you pointed out. Since overfitting is very important in ANN learning, as you mentioned, it was considered in the learning process. Table 5 shows the results of considering overfitting like this. Since we felt that the explanation about this was insufficient, we added lines 476 to 480 in Section 4.2. In addition, the training data, verification data, and test data do not overlap, and cross validation was performed to prevent learning from being biased toward specific data in the evaluation model. The explanation about this was added from line 437 to line 443 in Section 4.1.

7. What are the CPU performance or complexity requirements for IoT to still be able to predict leaked password in a given time.

Yes, we revised the part that you pointed out. Since the proposed evaluation model extracts the features regardless of the length or complexity of the password and predicts whether a leak has occurred, the time complexity is O(1). In this respect, it is suitable for low-performance IoT devices, and this is explained further from line 555 to line 558 in Section 4.2.

8. Table 5 and 6 Time (ms) column does not make sense if there is no reference to device that experiments were performed on (for example targeted IoT device).

Yes, we revised the part that you pointed out. We added a description of the environment in which the learning was carried out (IoT device, OS version, etc.) from line 469 to line 474.

Additionally, we reviewed the overall paper and improved the sentences and expressions. Thank you.

Reviewer 2 Report

The manuscript presents a new and improved password security prediction method that can determine whether a password has been leaked. Since such a prediction would require vast amounts of leaked password data to be collected and stored (which can be difficult in low-performance computing environments or terminals due to performance drop or lack of space) this method uses a security evaluation model that predicts whether a password selected by the user has been leaked by using deep learning. The deep learning security evaluation model provides protection of the leaked passwords database from outside attackers, since it is divided into a process in which the model learns by collecting the features of passwords and whether they have been leaked, and a process of predicting whether a password selected by a user has been leaked. The reliability of the model is confirmed with an experiment where authors compare its results with the results of predicting leakage from other security evaluation indicators.

The present form of the paper has several major issues:

  • The practical application of such a solution is vaguely described. It is mentioned that the solution is/could be used in an IoT device, but this choice has not been justified in detail.
  • The authors state that the proposed method can predict whether a password selected by the user “has been” leaked, which is not clear enough because of, at least, two reasons:
    • 1) if the method is meant to check whether the password has been leaked in the past, then why just not do that simple check in the database? I understand that it is more secure if the password is not sent to the server every time, but it somehow seems simpler to make that check and properly secure it, than to create a whole machine learning network
    • 2) if the method is meant to check whether there is a possibility that the password could be leaked in the future, then the phrase “could be leaked” should be used. If this is the case, then additional factors should be considered: passwords are leaked very often and whether a password will be leaked or not is merely a question of the security of the attacked system(s).
  • The main problem with the whole article is the approach. The authors build a neural network for data with just three columns. This is completely redundant, and nowhere in the experiment, the method is tested whether it is appropriate for such a problem. The method should be compared to other classic ML approaches: Random forest, Decision Tree classifier, naive Bayes etc. in order to justify why the neural network makes sense here. In fact, the presented results already suggest that it doesn’t make sense, because when the network has multiple layers (Table 5) and one layer has multiple neurons (Table 6), the results either don’t improve significantly or even get worse. This indicates that the neural network is inadequate.
  • A cross-validation of the method must be done, for the methodology to be sound.

Minor issues:

  • Proofreading of the paper is required – some sentences are not clear enough.

In summary, the contribution of the manuscript is questionable, the soundness of the presented technical method is arguable. Considering all these issues I recommend rejection.

Author Response

Thank you for your review.
We revised what you pointed out in the manuscript and summarized how and what to revise .

The manuscript presents a new and improved password security prediction method that can determine whether a password has been leaked. Since such a prediction would require vast amounts of leaked password data to be collected and stored (which can be difficult in low-performance computing environments or terminals due to performance drop or lack of space) this method uses a security evaluation model that predicts whether a password selected by the user has been leaked by using deep learning. The deep learning security evaluation model provides protection of the leaked passwords database from outside attackers, since it is divided into a process in which the model learns by collecting the features of passwords and whether they have been leaked, and a process of predicting whether a password selected by a user has been leaked. The reliability of the model is confirmed with an experiment where authors compare its results with the results of predicting leakage from other security evaluation indicators.

  • The practical application of such a solution is vaguely described. It is mentioned that the solution is/could be used in an IoT device, but this choice has not been justified in detail.

Yes, first of all, I would like to thank you for the review. As you pointed out, the explanation about the areas of the low-performance IoT environment to which the proposed evaluation model could be applied seems inadequate. By considering this, we added examples that can actually be used from line 152 to line 163 of Section 2.3.

  • The authors state that the proposed method can predict whether a password selected by the user “has been” leaked, which is not clear enough because of, at least, two reasons:
    • 1) if the method is meant to check whether the password has been leaked in the past, then why just not do that simple check in the database? I understand that it is more secure if the password is not sent to the server every time, but it somehow seems simpler to make that check and properly secure it, than to create a whole machine learning network
    • 2) if the method is meant to check whether there is a possibility that the password could be leaked in the future, then the phrase “could be leaked” should be used. If this is the case, then additional factors should be considered: passwords are leaked very often and whether a password will be leaked or not is merely a question of the security of the attacked system(s).

Yes, we revised the part that you pointed out.

1) In low-performance IoT environments, security mechanisms such as key encryption algorithms are limited due to low processing capacity and bandwidth. Therefore, there is a risk in sending the password selected by the user to the server to determine whether it has been leaked. This was reflected from line 164 to line 170 of Section 2.3. In addition, even if the proposed evaluation model is leaked in the IoT environment and exposed to a password attacker, the evaluation model that has been updated by continuously learning in the server is an ANN model with a weight that is completely different from the leaked evaluation model. Since this is updated in the IoT environment, the attacker cannot obtain valuable information from the leaked security evaluation model. This was reflected in lines 245 to 254 of Section 3.

2) This study presented the limitations of the existing methods to determine whether a password has been leaked from a server. To solve this, a security evaluation model that predicts leaked passwords in low-performance IoT environments was proposed. Since we felt that the argument about the need for such a study was inadequate, it was reflected from line 152 to line 170 in Section 2.3.

  • The main problem with the whole article is the approach. The authors build a neural network for data with just three columns. This is completely redundant, and nowhere in the experiment, the method is tested whether it is appropriate for such a problem. The method should be compared to other classic ML approaches: Random forest, Decision Tree classifier, naive Bayes etc. in order to justify why the neural network makes sense here. In fact, the presented results already suggest that it doesn’t make sense, because when the network has multiple layers (Table 5) and one layer has multiple neurons (Table 6), the results either don’t improve significantly or even get worse. This indicates that the neural network is inadequate.

Yes, we revised the part that you pointed out. Regarding the model that evaluates the security of passwords, the accuracy of the evaluation model, concealment of the learning process, and concealment of the learning result are important. Even if the evaluation model is leaked, a password attacker should not obtain any valid information from the leaked evaluation model. If this is considered, the decision tree is not suitable for password security evaluation models because the criterion for classifying each feature is not concealed. In addition, random forest is not suitable for training this evaluation model, which defines the number of features to be three. Therefore, ANN was used to train the evaluation model, and an explanation about this was added from line 437 to line 443. In addition, only three features were extracted to evaluate the security of passwords in low-performance IoT environments. The possibility of improvement by extracting password features that are a little more diverse was added from line 585 to line 589 in Section 5.

  • A cross-validation of the method must be done, for the methodology to be sound.

Yes, we revised the part that you pointed out. The proposed evaluation model used k-fold cross validation to prevent learning from being biased toward specific data. This was carried out for each epoch, and the results for the average accuracy and average loss values of the verification data are shown in Table 5 and Table 6. Since an explanation that cross-validation like this was performed seemed to be missing, it was added from line 429 to line 436 of Section 4.1.

  • Proofreading of the paper is required – some sentences are not clear enough.

Yes, we made the revisions that you asked for. We reviewed the overall paper and improved the sentences and expressions.

Thank you.

Reviewer 3 Report

The paper proposes a method to evaluate passwords’ security using a deep learning approach. The paper is interesting and generally well written. However, some issues need to be solved:

  1. English needs polishing. Here are some examples from abstracts: a) line 7 (first line of Abstract): it is written “have” instead of “has”; b) line 14: it is written “A deep” instead of “a deep”; c) line 15: it is written “using three feature data were extracted” instead of “using three feature data extracted”. Please check the entire manuscript;
  2. Line 20 (Keywords): there are three overlapping keywords: “Password security strength”; “Password”; “Password security”. I suggest to keep only “Password security strength” and instead of the other two to include “information security” and “authentication”;
  3. The sentence from lines 51-53 (“However, while vast…”) is badly constructed and needs to be rewritten.
  4. How the method proposed in this manuscript may be implemented on devices with low-computational capabilities? Some details are needed.
  5. Lines 80-85: Instead of “Section” the authors used “Chapter”. Please change.
  6. Line 95 and title of Figure 1: there is only one figure there. We do not need to notate Fig. 1a because we do not have Fig.1b for example;
  7. Line 219-220: the sentence “First the password…” is confusingly written and needs to be reshaped;
  8. Table 2: there is a typo (“times1”) that needs to be corrected;

Author Response

Thank you for your review.
We revised what you pointed out in the manuscript and summarized how and what to revise.

The paper proposes a method to evaluate passwords’ security using a deep learning approach. The paper is interesting and generally well written. However, some issues need to be solved:

1. English needs polishing. Here are some examples from abstracts: a) line 7 (first line of Abstract): it is written “have” instead of “has”; b) line 14: it is written “A deep” instead of “a deep”; c) line 15: it is written “using three feature data were extracted” instead of “using three feature data extracted”. Please check the entire manuscript;

Yes, first of all, I would like to thank you for the review. We made the revisions that you asked for and reviewed the overall paper and improved the sentences and expressions.

2. Line 20 (Keywords): there are three overlapping keywords: “Password security strength”; “Password”; “Password security”. I suggest to keep only “Password security strength” and instead of the other two to include “information security” and “authentication”;

Yes, we revised the part that you pointed out. Additionally, the keywords were arranged in alphabetical order.

3. The sentence from lines 51-53 (“However, while vast…”) is badly constructed and needs to be rewritten.

Yes, we revised the part that you pointed out.

4. How the method proposed in this manuscript may be implemented on devices with low-computational capabilities? Some details are needed.

Yes, we revised the part that you pointed out. First of all, since the explanation about the correlation between the proposed evaluation model and the low-performance IoT environment seems to be inadequate, lines 152 to 170 were added in Section 2.3. In addition, the reason for selecting the ANN model as an evaluation model was added from line 437 to line 443 in Section 4.1. Explanations about the experiment environment and IoT devices were added from line 469 to line 474 in Section 4.2. In conclusion, the argument that the proposed evaluation model can be used in low-performance IoT devices was added from line 555 to line 558.

5. Lines 80-85: Instead of “Section” the authors used “Chapter”. Please change.

Yes, we revised the part that you pointed out.

6. Line 95 and title of Figure 1: there is only one figure there. We do not need to notate Fig. 1a because we do not have Fig.1b for example;

Yes, we revised the part that you pointed out. In addition, the content written as Figure 1(a) in the text was all revised to Figure 1.

7. Line 219-220: the sentence “First the password…” is confusingly written and needs to be reshaped;

Yes, we revised the part that you pointed out.

8. Table 2: there is a typo (“times1”) that needs to be corrected;

Yes, we revised the part that you pointed out. Additionally, we reviewed the overall paper and improved the sentences and expressions. Thank you.

Round 2

Reviewer 2 Report

The authors included the recommendations and improved the quality of the paper.

Reviewer 3 Report

The authors have successfully solved all my comments and concerns. In my view, the manuscript was significantly improved.